# Designing Personas for E-Resources Users in the University Libraries

**Yuli Rohmiyati** [1,2], **Tengku Siti Meriam Tengku Wook** [1,*], **Noraidah Sahari** [1], **Siti Aishah Hanawi** [1] and **Faizan Qamar** [1]

1   Faculty of Information Science and Technology, Universiti Kebangsaan Malaysia, Bangi 43600, Selangor, Malaysia
2   Department of Library Science, Diponegoro University, Semarang 50275, Indonesia
*   Correspondence: tsmeriam@ukm.edu.my

**Abstract:** Persona is a method to create a user profile by describing a fictitious user through user experience. This persona study needs to be carried out for the benefit of system design according to the users' wishes because, so far, electronic resources (e-resources) are not widely used due to cognitive and affective factors such as limited subscription resources, limited user manuals, limited navigation features, and frequent errors when using electronic resources. This leaves the user feeling confused and stressed. The aim of this study is to obtain profiles of e-resource users in college libraries. The method used is an empathy map created with data from 32 users who answered questionnaires and participated in interviews. This study found four e-resource user personas in university libraries: lecturers, students, research assistants, and librarians. Users want a guide for using electronic resources that allows for chat and sharing, and which is fun and can be accessed from any device anytime and anywhere. The benefits of this study will be useful for designing e-resource systems according to users' wishes.

**Keywords:** persona method; user experience; e-resources user; university library





## 1. Introduction

University libraries aim to support education, research, and community service by providing various services related to information needs. Books, articles, journals, and other sources of information, especially electronic sources, are needed by lecturers or students to support the learning process. However, based on a previous study [1], electronic resources in university libraries still need to be improved. The cognitive and affective factors such as limited guidance, limited navigation features, and frequent errors when using electronic resources can make users feel negative emotions. Whereas emotions play an essential role in overcoming problems [2], this situation is undesirable because it hinders effective teaching and the quality of academic research [3], thereby reducing the professional productivity of university lecturers. This can also result in low university productivity [4].

Based on the problems above, this research will determine the interests and frustrations of e-resources users. The results of this research are in the form of e-resource user profiles in the libraries of universities. These profiles will then be used to design the e-resource system according to the wishes of e-resource users. The user profile will be built using the persona method.

Persona is a method that can communicate individual dimensions in a system, software, or product design [5]. The benefits of personas are confirmed by researchers in the field of human-computer interaction (HCI) [6–9]. As a method in user experience [10–12], personas help understand the point of view of users. So far, the end user has preconceived notions about what the user wants. End users assume that prejudice is a mental representation of users, arguing that data is taken from real data, whereas data about who users are,

why users use products, and what can be done in the future can cause friction between user segments and actual users. In addition, the outcome of the decision is not user-centered. Personas accurately describe reported user factual data and can reduce information gaps about users because they include user attributes and interests [13].

The challenge in adopting and using personas with qualitative methods is the accuracy of personas, because there is no agreed-upon metric by the researchers. Accuracy in the context of personas is defined as the average of the user's traits [14]. Personas are user-oriented techniques that analyse and express the objectives and requirements of various user types. Both qualitative and quantitative methods can be used to design personas. This includes collecting user data, segmenting and categorising data, and writing descriptions and narratives of personas. Data collection can consist of using existing data, and creating certain data through interviews, surveys, or questionnaires. Data are analysed quantitatively and qualitatively (e.g., latent semantic analysis, predictive models such as decision trees, or k-grouping mean). After the persona attributes are identified through research, a narrative is written. During this stage, the persona is given a name, a feature narrative description, a list of features, a photo, and possibly a basic document. Persona narratives can be used to validate personas, but the researcher can try to validate them further (for example, by checking members against data from interviews). It is important to note that the use of these methods is flexible, and their actual implementation may vary [15].

This study is based on persona research that has the potential to increase the rigour of persona design tools out of empathic concern for the user in Human-computer interaction. The structure of this article is as follows. An introduction to the persona, followed by an explanation of the framework and a discussion of the five design phases, followed by a brief section on framework usage. Finally, future work is discussed.

## 2. Literature Review

Personas represent groups of target users who share similar needs, traits, and objectives. As a part of the HCI user-centred design (UCD) process, this methodological concept was first introduced in the software development industry in the late 1990s and quickly gained popularity [11]. A persona, defined as a fictitious person representing a type of user [16], encapsulates an organisation's or software system's core users [14].

Persona profiles depicted as real people usually provide information about demographics, motivations, frustrations, and information specific to domains, for example, interest preferences [6]. Demographics are statistical data relating to specific populations or groups [17].

Various approaches can be used in persona design, including participatory design, value-sensitive design, reflective design, destruction design, and de-gendering design [8], as well as the Hierarchical Dirichlet Process (HDP), which is a topic model [18]. Another approach is to reduce subjectivity by utilising copious survey data (Behavior Risk Factor Surveillance System) to construct an empirically based persona framework and validate the outcomes using a series of ordered tests to show consistency [15].

Firdaus [18,19] proposed a persona-aware attention framework using an encoder-decoder approach for conversational agents. Firdaus explored ways to include the desired emotion [20] in responses to make them interactive and engaging and help make responses more human. Experimental results on the PersonaChat dataset show that the proposed framework outperforms the basic model and can generate interactive and emotional responses.

In order to help designers accurately represent older adults by evoking empathy, facilitating consideration of health problems and needs, and reducing reliance on stereotypes, Zhu [15] incorporated health personas for older adults into the design process. Zhu created a two-level quantitative methodology for constructing a persona framework from an unbalanced data set. Zhu contributed a strong method of persona construction to represent older adults.

Zaugg [21] identified and developed the undergraduate patron persona by involving communications students to develop a theory of the library patron persona using surveys, interviews, observation, and ethnographic methods. Ten personas (groups of user focus groups) who use various library services were identified in this study.

### 3. Framework

This section discusses the framework for designing e-resource user personas in a university library. Detailed guidelines were provided by Nielsen on using personas in research, stating that designers have no unanimous agreement about any method of creating personas or what a finished persona should look like [22]. The guidelines for user research and academic studies that used personas with groups of web users, were some sources from which this framework was adapted. The framework is divided into five phases adapted from previous methods for creating personas [23], as shown in Figure 1. Each phase clarifies the relationship between librarians and users. Previous research into persona design influenced the process of adapting into persona design [24]. The framework also refers to user research design guidelines for persona creation to ensure alignment with user research practice.

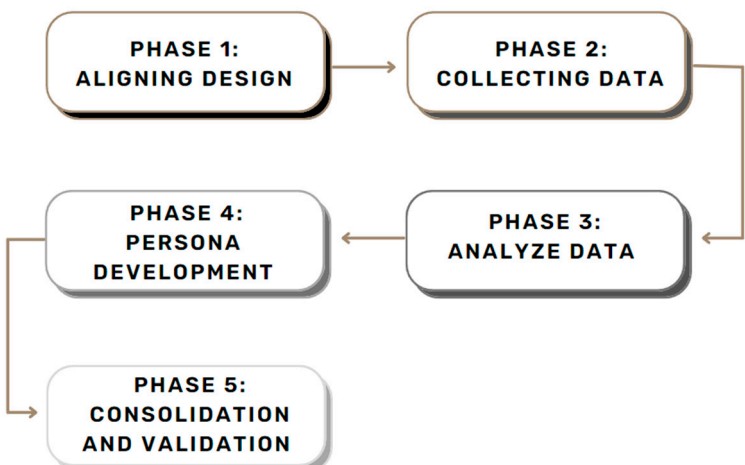

**Figure 1.** Framework Development.

The adapted framework is described in the following section. It describes the objectives for each phase and briefly mentions the framework's use. This framework illustrates data collection considerations for designing e-resources user personas in university libraries.

Phase 1: Aligning design goals and data collection strategy [25,26].

The design aims to mediate between libraries, in this case, librarians, and users of e-resources in university libraries.

Based on this study's objectives and available resources, the proper strategy for data collection should be used when interacting with users of e-resources in university libraries. The user research guide provides several potential ideas for defining a data collection strategy, which generally includes surveys, interviews, focus groups, and observation. However, many of the common data collection strategies described in this guide must consider design situations.

This study adopts the data collection strategy with an empathy map to accommodate better research involving user services, with consideration for the structure of the interview, focus group, or observation and how this might better accommodate users.

Phase 2: Collecting data from e-resource users (interviews and observations) [15].

Implementing the data collection strategy outlined in the previous phase is the goal of this phase. Data is usually collected through surveys, interviews, and observations, which involve e-resource users, but it can also include conversations with librarians. It is necessary to ensure that the data set is diverse and accurate to create a good persona, paying close attention to the questions asked in this phase.

This phase follows the previous guide on creating personas. Some standard questions for creating personas include user goals, demographics, skill sets, fears or pain points, technology experience, and expectations. Other questions used include questions about motivation and questions about social identity.

Specific questions in this research are related to user interests and ways to obtain information. Data from user tasks is used to determine user information-seeking behaviour.

Phase 3: Analyze data (user profiling) [27].

This phase aims to identify the user profile of e-resource users in the university library using the data collected in the previous phase. Data is analysed through appropriate methods in order to explore and determine relationships between collected data and detect general patterns. Statistical methods and approaches are typically used to analyse survey data, whereas thematic analysis is needed for qualitative data. Persona development based on theory should employ that theory to direct data analysis through the exploration of quantitative factors or through deductive thematic analysis and job status-specific coding.

Phase 4: Develop a persona description [22].

The goal of this phase is to use results from the data analysis to develop a series of user persona descriptions of e-resource users in the university library. Data is analysed based on the observed similarities and patterns. This process of grouping is known as segmentation. Each group will be used to develop a persona description following segmentation. Descriptions of personas should reveal information that allows designers to build empathy and understanding for the people they are designing for by knowing users' interests, goals, and frustrations.

Persona segmentation focuses on classifying the status of e-resource users. The user behaviour resulting from this phase is then used to describe important information related to the user's behaviour. The e-resource user persona descriptions can then be used to emphasise different service levels for each persona during the design process.

The type of information in an e-resource user persona should be guided by how it improves understanding of the librarian's relationship with e-resource users. Thus, the type of information about e-resource user personas should reflect the behaviour of the dominant user.

Phase 5: Consolidation, validation and persona description [22].

This phase focuses on consolidating the persona descriptions of e-resource users and validating them through feedback from end users. Persona adaptation, used for web users, emphasises the importance of feedback, and ensures consistency between personas and data.

The research context and stakeholders may change from time to time, and personas should be updated to reflect these changes. E-resource user personas should be updated based on ongoing interaction with librarians and e-resource users. Following an examination of each phase of the framework, it is necessary to consider briefly how the framework can be used in a design context. We argue that using these personas will allow university libraries to take a more holistic approach in designing e-resource systems. Policymakers must understand the needs of e-resource users in order to improve the library's service quality.

## 4. Results

The user demographics section includes questions about name, gender, occupation, age, education level, interests, country, and nickname. This question explains the user's background. Table 1 shows the demographic data of the respondents.

Respondents in this study consisted of 44 males and 56 females (Table 2). Based on their occupation, it is known that 22 of the respondents were students, 44 were lecturers, 12 were research assistants, and 22 were library assistants or librarians. As for the age of the respondents, 13 were 21–30 years old, 53 were 31–40 years old, and 34 were 41–50 years old. Based on the respondents' level of education, a total of 12 have a bachelor's degree, 66 have a master's degree, and 22 have a doctoral degree.

**Table 1.** Demographic data of respondents.

| Demographic | Category | Frequency (*n* = 32) | Percentage |
|---|---|---|---|
| Gender | Male | 14 | 44 |
| | Female | 18 | 56 |
| Occupation | Student | 7 | 22 |
| | Lecturer | 14 | 44 |
| | Research Assistant | 4 | 12 |
| | Staff | 7 | 22 |
| Age | 21–30 years | 4 | 13 |
| | 31–40 years | 17 | 53 |
| | 41–50 years | 11 | 34 |
| Level of education | Bachelor | 4 | 12 |
| | Master | 21 | 66 |
| | Doctoral | 7 | 22 |

**Table 2.** Country of respondents.

| Item | Frequency (*n* = 32) | Percentage |
|---|---|---|
| Malaysia | 8 | 25 |
| Indonesia | 7 | 22 |
| UK | 1 | 3 |
| Taiwan | 2 | 6 |
| Nigeria | 2 | 6 |
| Japan | 3 | 10 |
| Scotland | 1 | 3 |
| America | 2 | 6 |
| Germany | 1 | 3 |
| Austria | 1 | 3 |
| Australia | 1 | 3 |
| Netherlands | 2 | 6 |
| Singapore | 1 | 3 |

The respondents' country data are shown in Table 2.

Based on Table 3, 25 of the respondents were from Malaysia, 22 were from Indonesia, 10 were from Japan, 6 each from Taiwan, America, Nigeria, and the Netherlands, and 3 each from England, Scotland, Germany, Austria, Singapore, and Australia.

**Table 3.** User characteristics.

| User Characteristics | Requirement |
|---|---|
| Accessed electronic resources in university libraries. | Users have used electronic resources in university libraries. |
| Validly registered as a lecturer, student, or assistant. | Have a valid account to access electronic resources in the university's library. |
| Be able to use electronic resources in the university's library. | Users can use electronic resources well and understand how to use electronic resources. |

a.     Information media

Based on the questionnaire results, the information media that respondents often use are electronic resources in the library. As many as 53 of respondents often use e-resources from the library, 35 use google scholar, 6 use both the electronic resources in the library

and google scholar, and 6 use other media. The results for information media are shown in Figure 2.

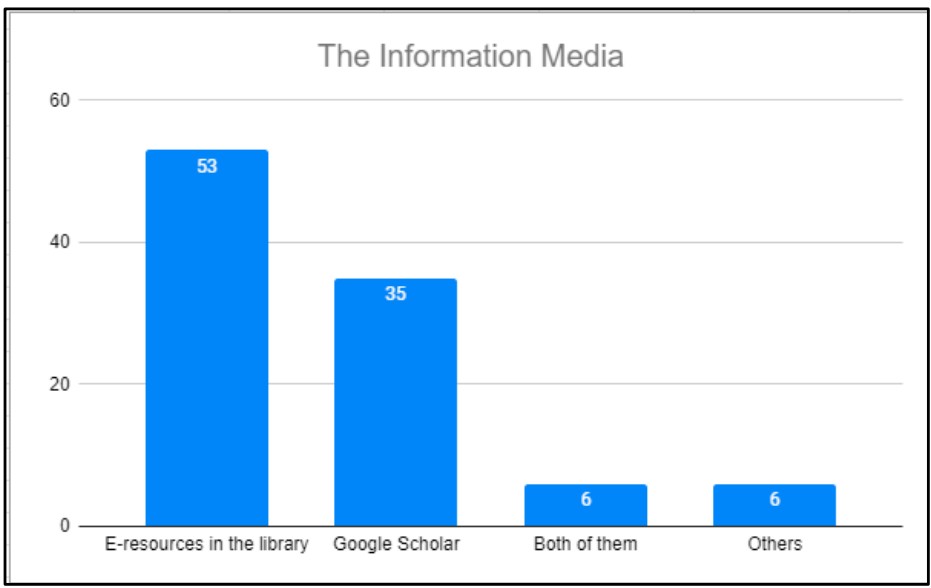

**Figure 2.** The Information Media.

The respondents who chose to use google scholar had the reasons of convenience (easy to search, easy to access, log in without conditions), speed of obtaining information, connection to various publishers, relevance, coverage of a broader range of scholarly works, ability to connect from anywhere, and plentiful options.

The respondents who chose electronic resources in the library had more complete and unrestricted access reasons. Electronic resources can retrieve high-ranking, new, scientific, reliable, and valid journals. It also allows access to current literature in various forms, is complete, and allows users to easily find the specific data needed.

Electronic resources in the university library have all the information needed for research or assignments. The information provided is also more diverse and sponsored by the university.

The library provides almost all the resources users need, such as technical articles from specific web searches such as WoS and Scopus. The university library will always inform students on how to access electronic resources in the library and announce all its services and features on its website and through all communication channels, such as e-mail and brochures.

Users of electronic resources who require a quick and general search usually use Google Scholar. But when the focus is on specific topics where the quality of references matter, electronic resources such as Scopusare preferred. The authors of this paper also frequently use electronic resources to access relevant scientific literature.

Moreover, some users use other electronic resources such as Libgen or SCIhub for ease of access compared with the university library. They prefer to type keywords in those public search engines and get the information they need easily and quickly.

b.    Technological Assistance

Based on the technological assistance required, 44 of respondents needed digital technology assistance, 34 needed smartphone assistance, and 22 needed other technological assistance (Figure 3).

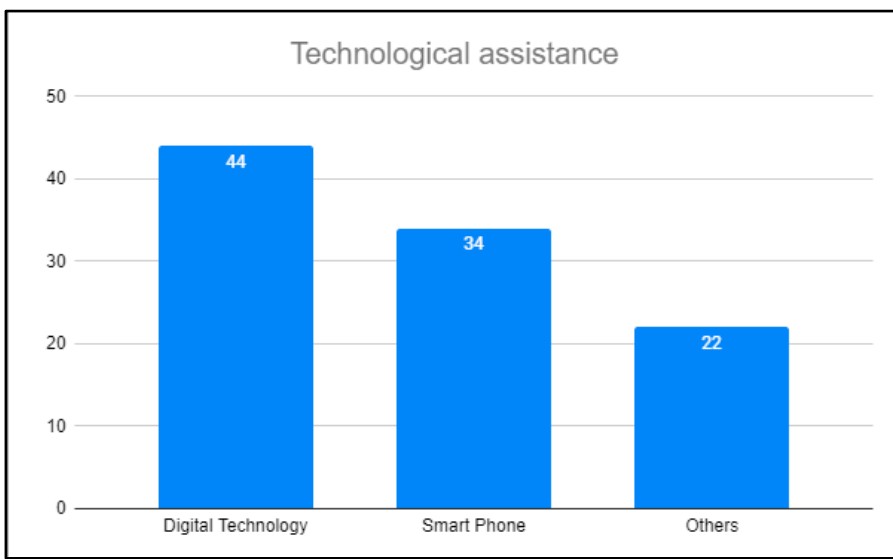

**Figure 3.** Technological Assistance.

Respondents who chose smartphones stated that they could find papers whenever they liked, for example, on the train, so it is more flexible, convenient, and can be used while travelling. Users are also happy if electronic resources are practical, available anywhere, and valuable. However, users also pointed out that the font on their smartphone screen is too small when accessing electronic resources through their smartphones.

As for the respondents who chose technological assistance in the form of digital technology, the reason given was that mobile applications on laptops or other devices such as smartwatches make their work easy. They need optimised search methods related to specific research, so they often use computer-based programs to help with research. They feel comfortable wih such devices and find it easy to access electronic resources. Smartphone screens have limited usefulness because they are too small for reading online journal articles. This interferes with the search for electronic resources. However, the respondents occasionally use their mobile phones to access electronic resources in university libraries.

Respondents feel they cannot solve the problem themselves if they get disconnected. Therefore, we need to help them because sometimes they need to ask about how to access electronic resources when they are not in campus, when using modern devices in classes, or when there are broadband-type compatibility and connectivity issues. The respondents also need formal digital technology training to access and easily use resources.

c.    Information Display Methods

Based on the information display method, 56 of respondents prefer information to be displayed in an interactive way, 28 prefer mobile applications, and 16 require other strategies. The results for information display methods are shown in Figure 4.

Respondents who chose the interactive display method stated that they would contact the librarian when they could not access the required material. Also, according to them, an interactive and intuitive user interface is preferred because it is easier to find specific articles. They hope there will be guidance or help to solve difficulties that arise during the search process.

The respondents believe interactive displays are more fun, friendly, attractive, and user-friendly, and allow them to collect information quickly. Whether it is a mobile application or a website, users need an interactive way to contact a librarian directly when they have a problem. They can directly contact the librarian if they encounter a unique situation not yet covered by the available instructions.

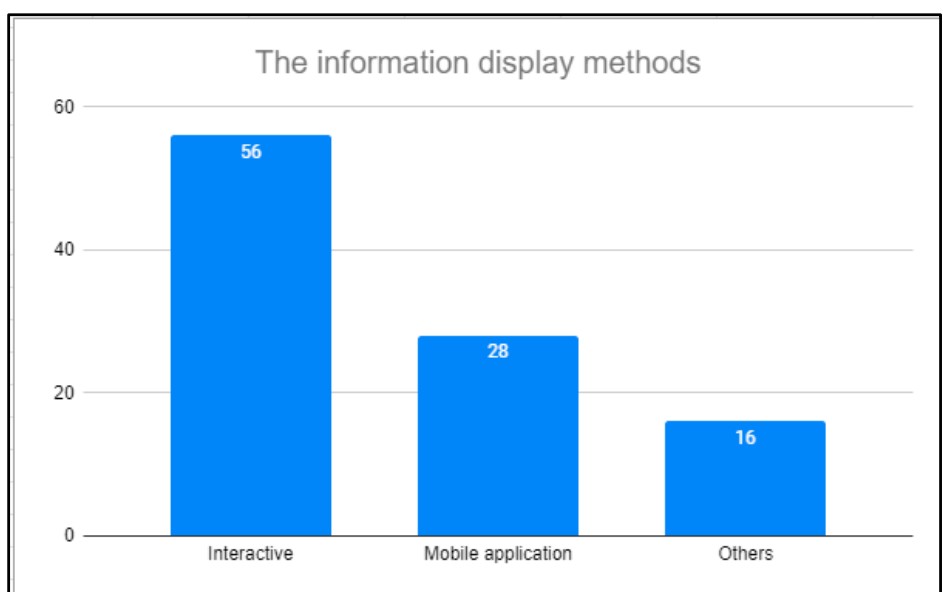

**Figure 4.** Information Display Methods.

As users, they need something different from just looking for references in the library and hope that each contact can be linked to an archive or other object owned by the university. This allows users to browse electronic resources more efficiently without consuming too much time, facilitating their activities in searching for information and facilitating communication.

As for the respondents who chose the method of displaying information in the form of mobile applications, the reasons were the ease of access, and that it is faster than having to open it from a laptop whenever they wanted to find materials. Proper network management facilitates problem-solving, provides access to electronic resources, and allows for flexibility in operation. This in turn, provides space for the exchange of ideas, which accelerates learning. Respondents felt that mobile apps are easier to carry and are highly accessible. However, having too many apps on display sometimes make them feel stressed.

d.    Reason for not Using Electronic Resources.

Based on the questionnaire results, it is known that 20 respondents do not use electronic resources in the library due to the lack of full text and lack of relevant material in some databases. A total of 14 respondents stated a lack of access to archival material, and 11 noted the inability to access databases from home and the lack of materials related to their subject (Figure 5).

Another reason for not using electronic resources is due to the need for more open access to other sources. Also, the system is too complicated with some new unfamiliar features, so the availability of professional librarians who can help users is required. The difficulty of obtaining the full text of a subscription journal makes users less personally motivated.

Sometimes users need links to collections from other universities, items not displayed in regular services, special collections, or paper quality filters such as Scopus or other high-quality publishers. Sometimes universities or institutions do not have such subscriptions, thus users have pay to access subscription-based materials. There is also less new information from journals and articles in the library.

Consumers usually have a limited time to find information from the library, even though they wanted to enjoy electronic resources that can be accessed from home.

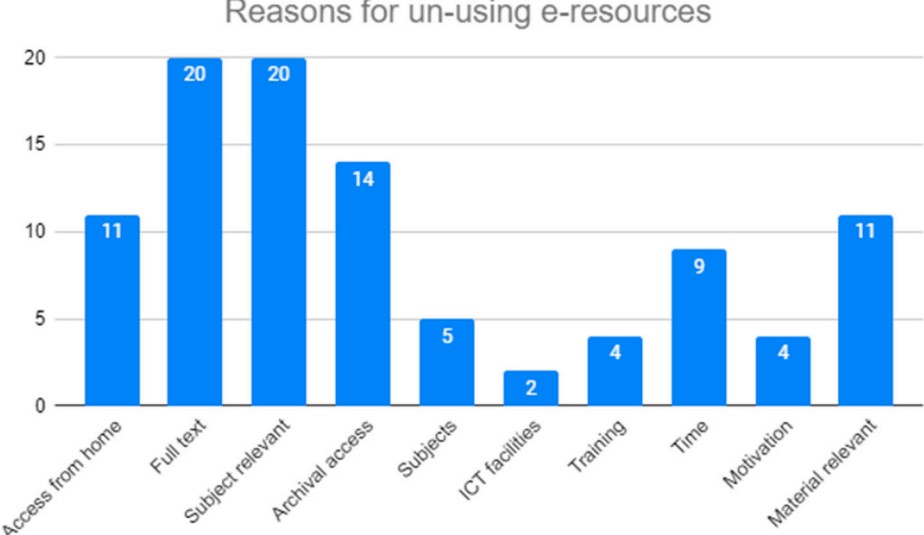

**Figure 5.** Reasons for not Using Electronic Resources.

e.    Problems accessing e-resources

Based on the questionnaire results, it is known that 17 lack subscriptions to more foreign journals, 12 stated that the system is too complicated to use, and lack the skills and knowledge to use electronic resources, and nine stated they lack expert help and support. The results for problems accessing e-resources is shown in Figure 6.

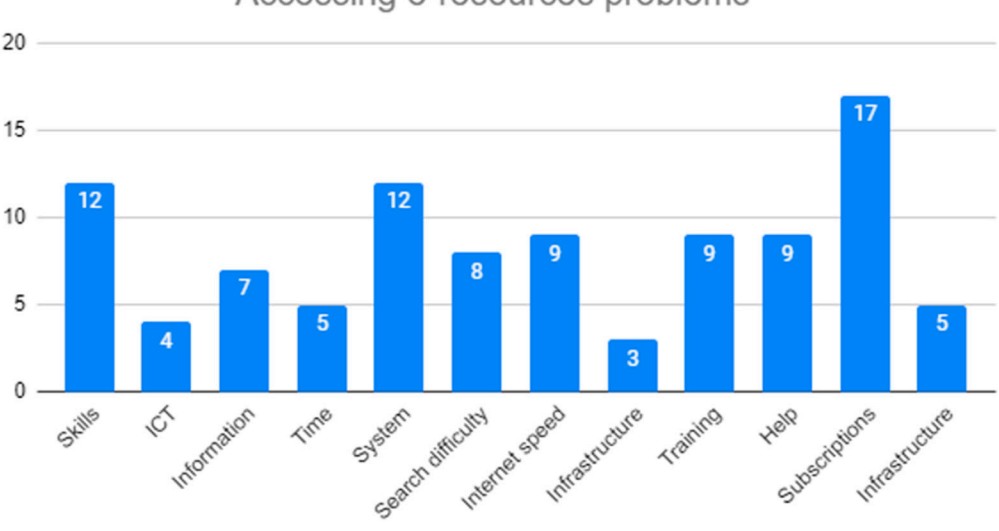

**Figure 6.** Problems Accessing E-Resources.

Another problem the respondents mentioned when using electronic resources is that sometimes it is impossible to get the full text, and the specific information needed is only occasionally available. Users expect a more organized database so that they are clear about every function on the website. Users also hope that other users could share their experiences using electronic resources in the university's library. Other problems mentioned by respondents are the lack of budget for developing electronic resources and the lack of personal information technology infrastructure.

## 5. Materials and Methods

Data collection for this section uses the empathic map method, which is doings, sayings, thoughts, and feelings of users. Data related to doings and sayings were collected

through user tasks, while data on thoughts and feelings were collected through questionnaire techniques. The questionnaire instrument was developed using an online google form application by sharing links to respondents selected through snowball sampling. A total of 40 respondents in various countries were contacted, but only 32 [27] gave feedback. The characteristics of users who are respondents are shown in the Table 3.

## 6. Discussion

Based on the data collection results above, a list can be formulated of user needs that have been identified through the items that users think, feel, say, and do,. In summary Table 4, the list consists of what technology the user needs, what kind of information display the user wants, and what type of experience [12,28] the user expects when using the university library's electronic resources.

**Table 4.** List of actual user needs that have been obtained and detailed through an online questionnaire.

| Technology | Information Display | Needs |
| --- | --- | --- |
| Digital Technology | Requires full-text information The information obtained can be | Fun with easy and quick access |
| An Interactive System | downloaded from a link | Shared experiences |

Users need digital technology that makes accessing these resources in university libraries easier. Users expect electronic resources to be accessed using personal computers, laptops, or other devices such as smartwatches. This technology is the primary tool for accessing electronic resources in university libraries. In addition, they also need an interactive system as a medium of information delivery. Users expect to be able to interact with librarians for assistance when users encounter problems while accessing electronic resources in university libraries. Technologies facilitating this interaction make users feel they are getting face-to-face services even when accessing online electronic resources.

The information display method that users want, i.e., full-text information that can be downloaded, requires a community that shares the experiences of other users when using electronic resources. This also helps other users who still need to become experts in using electronic resources in the university's library to learn from the shared experience.

Users also need the latest information that can add and expand knowledge. Users enjoy accessing electronic resources quickly without remembering or writing down usernames and passwords. Users also need a link to the archive or the library account.

*Personas*

Based on the empathy map that has been done, user profiles and personas can be developed. The complete personas are shown in Figure 7. There are 4 (four) categories of university libraty electronic resource users: lecturers, students, research assistants, and librarians. Shahnaz represents the lecturer, Abian represents the student, Cecilia represents the Research Assistant, and Mark represents the librarian. Each persona consists of at least 6–7 respondents.

(a)  Lecturers

This category is represented by education experts who want the latest knowledge for teaching and research. They are the people who are interested in reading full-text scientific articles in detail and can share their search results. These lecturers felt stressed and disappointed because it was challenging to get the full text; it was slow, and there were no links and no help from the librarian. This causes demotivation.

(b)  Students

This category is represented by users who like free services. If using an electronic resource, they want to be able to access it from any place, such as places outside the campus.

This group uses the latest technology, which is sometimes difficult to use. They need help to get the information they want with the advanced devices they have for their tasks.

(c)     Research Assistants

This category is represented by users who need high-impact articles. People in this group may feel like a failure if they are unable to get the relevant articles. This group requires up-to-date and detailed knowledge that can be downloaded. They need instructions to avoid getting confused when using the electronic resource system. They also need smartphone technology to access electronic resources on the go.

(d)     The Librarian

This category is represented by librarians, who identify as users, managers, and information providers. Librarians are professionals and want the latest knowledge to support work and work policies. They want to guide users, so they prefer smartphone technology so that it is easy to contact anyone who needs help from anywhere. Despite the lack of budget for developing electronic resources, and the occasional difficulty in getting articles because of the lack of university subscription, they still like to help other users.

**PERSONA**

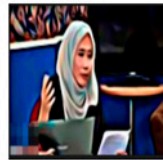 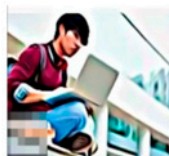 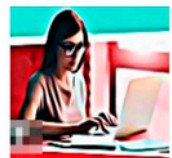 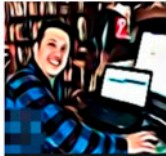

**Shahnaz The Lecturer**

"Done is better than perfect"

Age : 48

Education : Ph.D

Interest: Technology

Bio:

Shahnaz needs full-text articles for research and teaching. She needs expert support and likes digital technology with interactive information display methods. Her usual activities are browsing and downloading articles. She hopes to be able to get links. She is open to sharing, and for chat.

Frustation:

Slow access and no help

**Abian The Student**

" All our dreams can come true if we dare to pursue them"

Age: 22

Education: Bachelor

Interest: Modern devices

Bio :

Abian needs free articles for learning and research tasks with modern tools such as Digital technology with interactive information display methods. He wants to have a guide for using electronic resources or a workshop.

Frustration :

Paid articles and no full text

**Cecilia The Research Assistant**

"Do the best. "

Age : 32

Education: Master

Interest: Sightseeing

Bio :

Cecilia needs WoS-indexed article information for research that can be accessed through a smartphone with an interactive information display method, which can be quite fun when accessed from any device, anytime and anywhere.

Frustration :

Complex systems

**Mark The Librarian**

"Make things easier for others, and our affairs will be more accessible"

Age: 52

Education: Master's Degree

Interest: Reading

Bio:

Mark wants to help and guide users with effective methods despite his limited time. He also wants the University to subscribe to all subjects according to the university's knowledge. The only technology he needs is a smartphone with a mobile application information display to facilitate chatting with users.

Frustration: Lack of budget

**Figure 7.** Complete Personas.

## 7. Conclusions and Future Work

This study presents a framework highlighting the importance of increasing the use of electronic resources in university libraries. This framework explains how libraries, with their e-resources service systems, mediate user and stakeholder needs into persona creation.

This study describes the relationship between policymakers and users mediated by e-resources systems. Relationships are essential and should be considered explicitly in the design. This method can be used to investigate the values underlying persona creation and

whether it can be applied in other contexts. We recommend the exploration of personas and the value they can add to the design process, in future work, particularly in the context of designing personas to improve library services. This framework is part of ongoing research where the next step is to use personas in designing e-resource service systems in university libraries. Various types of interactive digital media can be applied, such as virtual reality, conversation agents, and robots. It will be essential for future designers to consider the relationship between stakeholders and users.

**Author Contributions:** Conceptualization, methodology and analysis, Y.R., T.S.M.T.W. and N.S.; writing—original draft preparation, Y.R.; writing—review and editing, T.S.M.T.W., N.S., S.A.H. and F.Q.; supervision, T.S.M.T.W., N.S., S.A.H. and F.Q.; funding acquisition, T.S.M.T.W. All authors have read and agreed to the published version of the manuscript.

**Funding:** The APC was funded by GP-2020-K009682, FTM2, FTM1, DAP, TAP K009682 Universiti Kebangsaan Malaysia.

**Data Availability Statement:** Not applicable.

**Conflicts of Interest:** The authors declare no conflict of interest.

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
