# Peer review of "Designing Personas for E-Resources Users in the University Libraries"

_computers, doi:10.3390/computers12030048_

Round 1
Reviewer 1 Report
-The authors could shorter describe the Persona method.
quickly gained popularity [11]. Persona, defined as a fictitious person representing a type 78
of user (Duda, 2018), encapsulates an organisation's or software system's core users [12]. 79
- (Duda, 2018). The authors should change this reference with a number [xx].
Table 2. Demographic data of respondents
-The authors presented an exact percentage. There are enough present numbers without decimal places after dots.
- Moreover, there is no need to write % near each number.
-The same deals with all numbers presented in tables, in the text, and the figures.
The authors should carefully correct references according to the Journal's requirements.
Keywords: I suppose that should be "Persona method" instead of "persona".
The introduction should highlight how the presented study deals with the main aims and scopes of the Computers journal.
Reviewer 2 Report
Interesting work, and well-studied. Methodology and Writing: relatively well written. Analysis is clear and focused. Below are minor suggestions:
1. I strongly suggest to mention appropriate methods for the phase «Phase 3: Analyze data» by writing their names to narrow down the scope of the approaches under consideration to the specific approaches used.
2. Captions inside figures make visible (ex. Fig.5, Fig.6, etc.).
3. The caption for figure 7 is placed on a separate page. The caption should be placed on the same side as the figure.
